# *Clostridioides difficile* Toxin B Induced Senescence: A New Pathologic Player for Colorectal Cancer?

**DOI:** 10.3390/ijms24098155

**Published:** 2023-05-02

**Authors:** Katia Fettucciari, Alessandro Fruganti, Fabrizio Stracci, Andrea Spaterna, Pierfrancesco Marconi, Gabrio Bassotti

**Affiliations:** 1Biosciences & Medical Embryology Section, Department of Medicine and Surgery, University of Perugia, 06129 Perugia, Italy; pierfrancesco.marconi@outlook.it; 2School of Biosciences and Veterinary Medicine, University of Camerino, 62024 Matelica, Italy; 3Public Health Section, Department of Medicine and Surgery, University of Perugia, 06129 Perugia, Italy; fabrizio.stracci@unipg.it; 4Gastroenterology, Hepatology & Digestive Endoscopy Section, Department of Medicine and Surgery, University of Perugia, 06129 Perugia, Italy; gabassot@tin.it; 5Gastroenterology & Hepatology Unit, Santa Maria Della Misericordia Hospital, 06129 Perugia, Italy

**Keywords:** *Clostridioides difficile* toxin B, *Clostridioides difficile* infection, cellular senescence, senescence-associated secretory phenotype, irritable bowel syndrome, inflammatory bowel diseases, colorectal cancer

## Abstract

*Clostridioides difficile* (*C. difficile*) is responsible for a high percentage of gastrointestinal infections and its pathological activity is due to toxins A and B. *C. difficile* infection (CDI) is increasing worldwide due to the unstoppable spread of *C. difficile* in the anthropized environment and the progressive human colonization. The ability of *C. difficile* toxin B to induce senescent cells and the direct correlation between CDI, irritable bowel syndrome (IBS), and inflammatory bowel diseases (IBD) could cause an accumulation of senescent cells with important functional consequences. Furthermore, these senescent cells characterized by long survival could push pre-neoplastic cells originating in the colon towards the complete neoplastic transformation in colorectal cancer (CRC) by the senescence-associated secretory phenotype (SASP). Pre-neoplastic cells could appear as a result of various pro-carcinogenic events, among which, are infections with bacteria that produce genotoxins that generate cells with high genetic instability. Therefore, subjects who develop IBS and/or IBD after CDI should be monitored, especially if they then have further CDI relapses, waiting for the availability of senolytic and anti-SASP therapies to resolve the pro-carcinogenic risk due to accumulation of senescent cells after CDI followed by IBS and/or IBD.

## 1. Introduction

*Clostridioides difficile* (*formerly Clostridium difficile*, *C. difficile*) [1,2] is a Gram-positive endospore-forming anaerobic bacteria that is the causative agent of infectious diarrhea, ranging from mild symptoms to pseudomembranous colitis and toxic megacolon, a transmural pan-colitis that may require colectomy due to life-threatening fulminant infection [3,4]. *C. difficile* produces three different toxins, toxin A (TcdA), toxin B (TcdB), and a binary toxin called *C. difficile* transferase (CDT) [5]. TcdA and TcdB are primarily responsible for the pathogenic action of *C. difficile*, with TcdB being more toxic than TcdA and responsible for the major effects of *C. difficile* infection (CDI) [5,6,7,8,9]. In fact, TcdA and TcdB, after binding and internalization, induce in all cell types studied the glucosylation and inactivation of Rho-GTPase which in turn leads to cytopathic effects (early cytoskeleton disruption, cell rounding, cell cycle arrest) and cytotoxic effects by apoptosis or necrosis, and production/secretion of chemokines/proinflammatory cytokines [5,6,7,8,9,10]. Most importantly, Fettucciari et al. demonstrated that enteric glial cells (EGCs) surviving TcdB-induced apoptosis [10] become senescent as a survival response to TcdB stressor stimulus [11].

Cellular senescence is a fundamental stress response of all somatic cell types to a wide range of stressors [12,13]. The most studied inducers of senescence are: anticancer drugs, radiation, tissue repair processes, bacterial infections (mainly through the action of toxins), and more recently, viral infections [14,15]. The main effectors of the cellular senescence program are activated oncogenes, cytokines, reactive oxygen species (ROS), and persistent DNA damage [15,16,17,18]. Cells that acquire the senescence state undergo global changes that profoundly influence their phenotype and functions [15,19]. The senescent cells are thus characterized by irreversible cell cycle arrest, changes in cell metabolism, morphology, and gene expression, an increase of senescence-associated beta-galactosidase activity (SA-β-Gal), early and persistent DNA damage response (DDR), senescence-associated heterochromatin foci (SAHF), and senescence-associated secretory phenotype (SASP) [19,20]. The main changes that senescent cells undergo have profound implications on their physiological and pathological role [21]. Moreover, some of these changes also become senescence markers [22].

It is worth noting that TcdB induces senescence in EGCs that survive the action of the toxin [11], which predominantly induces death by apoptosis/necrosis [10,23]. TcdB-induced senescence in EGCs mainly triggers irreversible cell cycle arrest and persistent DNA damage and downregulation of c-myc and hypophosphorylation of phosphorylated retinoblastoma protein (pRb). TcdB-induced senescent cells undergo cell cycle arrest in G0/G1 and G2/M mediated by overexpression of p27 associated with the downregulation of cyclin-dependent kinase 1 (CDK1) and cyclin B1 [11]. Furthermore, the sirtuins seem to contribute to the maturation, maintenance, and stabilization of the TcdB-induced senescence state, by antagonizing the lethal effects of high levels of ROS through an antioxidant response, thus favoring the survival of senescent cells [11].

Until now, there is no in vivo demonstration of cellular senescence during CDI. However, studies on the effects of *C. difficile* toxins (Tcds) in the tissue of mice during CDI demonstrated changes in the expression of cell cycle inhibitors and cyclin-dependent kinase inhibitors (CDKIs) indicative of cell cycle arrest [24,25] strongly suggesting that cellular senescence could occur during CDI in vivo.

Cellular senescence has been reported also in several gastrointestinal diseases including irritable bowel syndrome (IBS), inflammatory bowel diseases (IBD), and colorectal carcinogenesis [26,27,28]. In subjects with IBS, indeed, the persistent low-grade inflammation may induce senescent cells. In subjects with IBD, there is a pathological stress of gastrointestinal cells due to the strong and persistent inflammatory response and to a strongly altered microbiome which contributes to the inflammatory state and continuous processes of tissue repair which eventually lead to the formation of senescent cells [27,28,29]. Of interest, IBD patients are at risk for the development of CDI and have a substantial increased risk of colorectal cancer (CRC) [30,31,32,33,34,35].

CRC, which includes colon and/or rectum cancer, is a significant problem of health as is the third most commonly diagnosed cancer and cause of cancer-related deaths in the United States (US) [36]. However, it ranks second in cancer-related deaths globally [37] and is the leading cause in men younger than 50 years [36,38]. Indeed, in 2023, approximately 153,020 individuals will be diagnosed with CRC and 52,550 will die from the disease [37,39], including 19,550 cases and 3750 deaths in individuals younger than 50 years [36,38]. The new treatments have improved outcomes, but the five-year relative survival remains poor at only 64.7% [39,40]. The odds are worse for patients with metastatic disease as their five-year survival rate is less than 20% [39]. CRC is caused by aberrant proliferation of the colon’s glandular epithelial cells. Three principal types of CRC have been described: sporadic, hereditary, and colitis-associated [41]. Both hereditary and environmental risk factors are involved in the development of CRC. The environmental risk factors of about one-half of all cases and deaths are due to smoking, an unhealthy diet, high alcohol consumption, physical inactivity, excess body weight, and chronic inflammations including ulcerative colitis and Crohn’s [36,39,40,41,42]. In addition, a large proportion of CRC incidence and mortality is preventable through the receipt of regular screening, surveillance, and high-quality treatment [36,37,39].

The demonstration that TcdB induced in vitro cellular senescence is of considerable importance given the increasing diffusion of CDI, the frequent recurrences, and the correlations between CDI and IBS and IBD. Therefore, whether senescent cells are generated during CDI, the pathological effects of cellular senescence related to persistent inflammation could contribute to the development of IBS and IBD, immune suppression, and finally tumor induction. Of interest, epidemiological data of senescence induced by bacterial genotoxins associate it with a higher incidence of IBD and CRC [43,44,45,46,47,48].

Senescent cells are acquiring an increasingly important role both in the pathology and prognosis of CRC. Indeed, it has been reported that in patients with CRC, a high number of senescent cells are associated with a poor prognosis, underlining how a high concentration of senescent cells favors tumor progression. The senescent cells initially can play a protective anti-tumor effect, but this can become pro-carcinogenic over time and with changes in the microenvironment conditions [27,29,31,49,50,51,52].

Therefore, the induction of senescent cells by *C. difficile* TcdB during CDI, the observation that CDI occurs mainly in aging people where there is a physiological increase in senescent cells for the advancing age and the induction of cellular senescence in IBS and IBD could cause an accumulation of senescent cells, especially in subjects who develop IBS and/or IBD after CDI. This strong accumulation of senescent cells with long survival can alter the balance in the anti- pro-tumoral effects and favor neoplastic progression, pushing pre-neoplastic cells towards the complete neoplastic transformation in CRC by the SASP. Pre-neoplastic cells could appear in the colon as a result of various pro-carcinogenic events, such as infections with bacteria that produce genotoxins which generate cells with high genetic instability [53,54,55,56].

In this review, we highlight and discuss some important aspects that suggest, albeit indirectly, that CDI and its recurrences by induction and accumulation of senescent cells in the gastrointestinal tract can represent a risk for the induction of CRC above all if after CDI occurs IBS and/or IBD.

This review described for the first time and highlights the possibility that in subjects developing IBS and/or IBD after CDI and its recurrences CDI by induction and accumulation of senescent cells could be a potential factor to develop CRC.

## 2. CDI

### 2.1. C. difficile Epidemiology

CDI often occurs after antibiotic treatment in aged people (>65 years; median age 72) or after hospitalization defined as “hospitalization associated CDI” [57,58], but also in young individuals considered at low-risk (<65 years; median age 50–51), healthy, not exposed to an antibiotic, hospitalizations, and healthcare interventions [59,60,61]. The CDI induced in these people is defined as “community-acquired CDI” and is increasing worldwide. However, there are several other predisposing factors that favor CDI [9,60,62], such as the use of drugs (e.g., proton-pump inhibitors) [63], types of diets [9], the genetic and immune system of individuals [64] and comorbidities as diabetes [65], hypoalbuminemia [66], obesity [67], chronic kidney disease [68], IBD [69,70,71,72,73,74] and impairment of immune system [73,75]. Of note, it has been reported that *C. difficile* colonizes about 4–15% of healthy adults [76,77] and that in healthy infants the overall *C. difficile* colonization prevalence was 33.7% and the colonization rate by a toxinogenic strain was 7.1% [78].

The incidence, severity, and associated mortality of CDI worldwide, especially in elderly patients, are progressively increasing in recent decades [79] and the Centre for Disease Control and Prevention in 2019 cataloged CDI as one of the top three urgent antibiotic resistance threats [3,59,61]. Globally, the incidence of CDI is 2.2 per 1000 hospitalizations per year and 3.5 per 10,000 patient-days per year. Incidence rates per 10,000 patient days range from about 3 in France, Italy, and the US, to 6–7 in Germany, Poland, and Canada, with similar results provided by other studies [80,81,82]. In the US, CDI is associated with about 500,000 cases and 15,000–30,000 deaths/year, and in Europe with about 124,000 cases/year with an overall mortality of 3–30% [62,82,83]. Overall, the mortality related to CDI is about 5% and it increases in high-risk groups such as elderly hospitalized and patients with comorbidities [82,83,84,85,86]. CDI represents more than 30–40% of healthcare-associated gastrointestinal infections [57]. However, these values likely underestimate the true incident cases [82].

This new epidemiological framework has been associated with the emergence of *C. difficile* strains with traits of high virulence and multidrug resistance to several classes of antibiotics, which present local or international diffusion. Recently, the highly virulent epidemic ribotype 027 has been globally recognized as a primary cause of CDI [87,88]. Numerous other epidemic ribotypes with high virulence traits are increasingly emerging, some at an international level, such as ribotype 078, ribotype 017, and ribotype 106 [88,89,90,91,92,93,94], and others at the national level, such as ribotype 018 and ribotype 607 (also known as ribotype 356) in Italy [81,95]. Importantly, CDIs induced by highly virulent strains are often coupled with a higher number of colitis needing colectomies, and higher mortality [96,97], with a rise in hospitalization. It is worth noting that highly virulent epidemic ribotypes in particular ribotype 027 and ribotype 078 (identified by rRNA-based phylogenic analyses) are associated with a higher number of relapses. The recurrence of CDI represents one of the most important challenges in the management of patients with CDI, causing a strong impact on patient quality of life, re-hospitalization rate, morbidity, and mortality. European Centre of Disease Control and Prevention, with surveillance conducted in 2016 in 556 hospitals in 20 European countries, estimated that 15 to 35% of patients with CDI relapsed within 4–8 weeks of discontinuing therapy antibiotics and that in 20.7% of cases, the infection had a fatal outcome [98,99,100,101].

In patients undergoing CDI, the relapse is usually due to the unsuccessful eradication of the infecting strain or to reinfection due to a different strain, the most frequent event [98,100,101].

CDI is assuming the characteristics of a global endemic spread, that cannot be arrested for the following reason [102]: (1) the presence of *C. difficile* in an anthropized environment (i.e., foods, products, and farm and domesticated animals) [103]; (2) the strong resistance of *C. difficile* endospores to the external milieu that, together with the complex interaction of *C. difficile* with the host, will further promote its colonization [104]; (3) the ability of *C. difficile* to make an infected person a strong spreader of the infection; (4) emergence of *C. difficile* strains which have increased rate of colonization, morbidity, and mortality, as the hypervirulent ribotype 027 and ribotype 078, producing higher amount of Tcds, Tcd variants, and/or producing a CDT [87,105]; (5) increasing ability of colonization of human hosts favored by dysmicrobism and inflammation [102]. In fact, dysmicrobism favors the susceptibility to infection and *C. difficile* overgrowth while inflammation, increasing cytotoxic activity of low doses of TcdA and TcdB could damage the colon microenvironment and enhance *C. difficile* colonization and infection.

### 2.2. Mechanisms of C. difficile Pathogenesis: In Vitro and In Vivo Effects

*C. difficile* produces three different toxins, TcdA, TcdB, and a binary toxin called *C.* CDT [5,6,9] TcdA and TcdB are primarily responsible for the pathogenic action of *C. difficile*, with TcdB not only being more toxic than TcdA but appearing to be responsible for the major effects of CDI [5,6,7,8,9]. While CDT is unable to cause disease by itself, when combined with the other two Tcds it appears to contribute to more severe disease [6].

The TcdA and TcdB bind to the Tcd receptors in the cell membrane and induce its internalization by an endocytic vacuole in which the acid pH promotes Tcd conformational modifications that favor the insertion of Tcd catalytic domain outside the membrane for cleavage, and the subsequent activation by glucosylation of the catalytic site of Rho-GTPase to inhibit their activity [5,6,9]. Glucosylation of Rho-GTPase leads to various biological effects such as [5,6,7,8,10] (Figure 1 and Figure 2): (1) early cytoskeleton disruption and cell rounding; (2) cell cycle arrest; (3) and cell death, which occurs after the cycle arrest, mainly by necrosis and apoptosis. Cytotoxic activity by apoptosis and necrosis by TcdA and TcdB has been described in several cell types present or recruited in the site of CDI [5,6,7,8,9,10]: colonocytes [106], enterocytes [107], myocytes [108], enteric neurons [109], EGCs [10], immune cells [110,111,112], but also in cells not present in the site of CDI such as hepatic cells [113], cardiac cells [114], lung fibroblasts [115], and nervous cells [116].

TcdB-induced necrosis shows early plasma membrane permeability demise, cellular leakage, ATP depletion, and chromatin condensation without caspase activation [5,6]. TcdB induces necrosis by strong production of ROS which causes necrosis likely by protein oxidation, lipid peroxidation, DNA damage, and/or mitochondrial dysfunction (Figure 1) [6,117]. On the contrary, TcdA at all concentrations used does not stimulate ROS production and it induces mainly apoptosis [5,6].

Especially for TcdB, the type of cell death induced is strictly dependent on the dose of TcdB and by TcdB receptors involved. In fact, TcdB at lower concentrations binds to the Chondroitin sulfate proteoglycan 4 or Frizzled receptors causing apoptosis in a glucosylation-dependent way. At higher concentrations (100 pM or above) binds to Chondroitin sulfate proteoglycan 4, Frizzled receptors, or poliovirus receptor-like 3 inducing necrosis in a glucosylation- and autoprocessing-independent way [118,119,120,121,122].

Apoptosis induced by TcdA and TcdB in several cell types has been widely studied and occurs mainly by caspase-dependent pathways mediated by extracellular and mitochondrial signaling [5,6,7,8,10,23,123].

However, TcdA and TcdB-induced apoptosis occurs also in a caspase-independent manner. In fact, Notrott et al. showed that TcdA induces apoptosis in HT-29 cells by activating not only caspases but also cathepsins and calpains [124]. Moreover, Fettucciari et al. in 2022 [23] demonstrated that TcdB induces apoptosis in EGCs in a more complex manner than previously thought (Figure 1) [10,23]. Indeed, TcdB activates three signaling pathways mediated by calpains, caspases, and cathepsins, which are all involved in both the triggering and effector phases of apoptosis signaling [23]. Calpain activation induced by the influx of Ca^2+^ is the initial pro-apoptotic signal in TcdB-induced EGC apoptosis and induces the majority of apoptosis by caspase-3, caspase-7, and poly ADP-ribose polymerase activation [23]. The latter is activated also by the initiator caspase-8, caspase-3, and caspase-7 [23]. Moreover, cathepsin B contributes to triggering the pro-apoptotic signal and contributes to one-third of apoptosis in a caspase-independent manner [23].

Fettucciari et al. also showed that pro-inflammatory cytokines, tumor necrosis factor-alpha (TNF-α) plus interferon-gamma (IFN-γ) (CKs) strongly enhanced TcdB-induced apoptosis [10,23] by increasing the activation of the three apoptotic pathways each mediated by calpains, caspases, and cathepsins (Figure 1) [23] and this phenomenon could have an important implication on CDI pathogenesis, CDI relapses, IBS and IBD [72,73,74,125,126].

Of note, TcdB can induce cell death also by (Figure 1): (1) pyroptosis, a type of cell death which causes strong inflammation, featured by pore formation and cell swelling/lysis with release of interleukin-1beta (IL-1β) and interleukin-18 (IL-18) after caspase-1-dependent activation of gasdermin D and processing of pro-IL-1β and IL-18, (2) pyknotic cell death, typified by chromatin condensation and ballooning of the nuclear envelope that is glucosyltransferase domain-dependent and -independent and ROS-independent, and (3) autophagy [5,6,7,8].

The demonstration that Tcds induces different types of cell death which can be triggered by different independent or overlapping pathways is a very important and peculiar aspect of *C. difficile* because different pro-apoptotic signal transduction pathways or cell death types can have distinct outcomes in the pathogenesis of CDI (Figure 1) [23].

Most importantly, Fettucciari et al. demonstrated that EGCs surviving the apoptotic TcdB effects become senescent as a survival response to a stressor stimulus triggered by TcdB (Figure 1 and Figure 2) [10,11].

The most evident effects of Tcds in vivo are a loss of tight junction, loss of intestinal membrane integrity, and above all cytotoxicity, leading to the destruction of the colonocyte barrier (Figure 1 and Figure 2) [6,8,127]. Regarding disruption of tight junction by Tcds it has been reported that Tcds increases paracellular permeability in colonic epithelial cells by mechanisms involving Rho-GTPase glucosylation and actin depolymerization, but also by direct effects on tight junction proteins [128,129]. Regarding the mechanisms of the increase in paracellular permeability induced by TcdA and TcdB involving RhoA glucosylation and actin depolymerization, it has been reported that is associated with disorganization of apical and basal F-actin [129,130]. F-actin restructuring was paralleled by the dissociation of occludin, Zonula occludens (ZO)-1, and ZO-2 from the lateral tight junction membrane without influencing the subjacent adherens junction protein, E-cadherin; there is also a decreased association of actin with the tight junction cytoplasmic plaque protein ZO-1, then Tcds modulates both the localization of tight junction proteins and the affiliation of tight junctions with the underlying actin cytoskeleton [129,130]. However, because Tcds-mediated decline in transepithelial electrical resistance preceded changes in cell morphology and tight junction ultrastructure this indicates that early cellular responses that occur before Rho-GTPase glucosylation contribute to the Tcds-induced permeability changes by direct effects of tight junction proteins [129,131]. In particular, it has been reported that TcdA stimulated the activities of membrane and cytosolic protein kinase C α and cytosolic protein kinase C β, and protein kinase C signaling regulate TcdA-mediated transepithelial electrical resistance, paracellular permeability changes and increased translocation of ZO-1 from tight junction [131]. However, TcdA did not induce translocation of ZO-2, dephosphorylation, or translocation of occludin [131]. Recently, it has been demonstrated that the decrease of tight junction proteins and disruption of gut integrity during CDI is also mediated by *C. difficile*-induced downregulation of peroxisome proliferator-activated receptor-γ in colonic epithelial cells [132].

Of course, in vivo, the picture of CDI is aggravated by the strong inflammatory response and the recruitment of innate immunity cells (Figure 1 and Figure 2) [5,6,133,134]. Immune cell recruitment is induced indirectly by Tcds-induced tight junction dysfunction but also by Tcds-induced secretion of pro-inflammatory cytokines from epithelial cells and from cells of the underlying lamina propria through activation of nuclear factor kappa-light-chain-enhancer of activated B cells, activator protein 1 and inflammasome [5,6,135]. In fact, overall, Tcds stimulate epithelial cells to release inflammatory mediators that recruit neutrophils to the colonic mucosa. Tcds after the disruption of the epithelial barrier penetrate the lamina propria and directly stimulate immune cells (dendritic cells, macrophages, mast cells) to release pro-inflammatory mediators, which amplify neutrophil recruitment and inflammation [6]. Tcds activates also enteric neurons and enhanced the production of substance P which induces mast cell degranulation and can stimulate macrophages of the lamina propria to release pro-inflammatory cytokines. Therefore, prolonged intestinal inflammation can amplify tissue damage and contribute to neutrophil infiltration into the lumen [6]. Additionally, CDT, expressed by some *C. difficile* strains, can act synergistically with Tcds to enhance pro-inflammatory cytokine secretion by innate immune cells [6].

In all the above contexts of diffusion, *C. difficile* increases the number of infected and sick subjects in which the cytotoxic action of TcdB not only causes intense cell death with the pathways described above but, in subjects who recover from the infection, increases also the number of senescent cells induced by the TcdB and, as we shall see, the processes of repair (Figure 2). All these events promote recurrences and pave the way for conditions such as IBS, IBD, and also CRC.

## 3. Cellular Senescence

Cellular senescence is a fundamental stress response by all somatic cell types to a wide range of stressors characterized by irreversible and persistent cell cycle arrest, changes in cell morphology, gene expression, and cell metabolism [136]. The most studied inducers of senescence are: anticancer drugs, radiation, tissue repair processes, bacterial infections (mainly through the action of toxins), and more recently, viral infections [14,15,16,17,22,136,137].

The main effectors of the cellular senescence program are activated oncogenes, cytokines, ROS, and persistent DNA damage [15,16,17,18,138]. Cells that acquire the senescence state undergo global changes that profoundly influence their phenotype and functions [15,19]. The main changes that senescent cells undergo have profound implications on their physiological and pathological role [15,17,139]. Moreover, some of these changes also assume the role of senescence markers [15,22,139] although only a combination of many markers will permit the detection of senescent cells in tissue in vivo [140,141]. These main changes are:Exit from the cell cycle with permanent cell cycle arrest due mainly to the activation of cell cycle inhibitors [p53, pRb, phosphatase, and tensin homolog (PTEN) and c-myc] which causes the activation of CDKIs such as p21, p16, p15, and p27. The latter blocks the formation of the CDK-cyclin complexes involved in the control of the cell cycle checkpoints that regulate the G1-S or G2/M phase transition. The CDKIs (p21, p16, p27) are the key molecules of the main classical senescence-inducing pathways p53/p21/pRb, p16/pRb, and PTEN/p27 [14,15,16,17,19];DDR activation: the persistence of DNA damage is a strong activator of the DDR and of senescence [16,17,18];Positivity for SA-β-Gal due to the increase of activity of lysosomal enzymes;Cellular morphology: senescent cells assume a flat and large morphology;Biological properties: senescent cells are more resistant to apoptosis induced by several stimuli;SASP: the acquisition by the cell of a SASP characterized by the secretion of important molecules that can be soluble [19,20] but also be localized in extracellular vesicles divided into, microvesicles and exosomes [142,143,144,145], represents the most relevant aspect of the senescence state. In fact, through the soluble SASP, senescent cells influence the cells of the surrounding microenvironment through a paracrine action that can be accompanied by juxtacrine and autocrine actions [16,17,22,139,146]. Bacterial components originating from bacterial microbiota such as damage-associated molecular patterns, pathogen-associated molecular patterns, and lipoteichoic acid that activate natural immune responses can induce or contribute to SASP. Moreover, bacteria, and especially some of their toxins, are inducers of cellular senescence and therefore of SASP [54,55,147].

The senescent cells can have profoundly different purposes and results because they are involved in the regulation of development, tissue homeostasis, in wound healing where myofibroblasts, which under severe stress undergo an acute form of senescence, limit the size of the scars, i.e., the hyperproliferation of the repair tissue, after providing support to the recovery of parenchymal cellularity. Furthermore, senescent cells, even those originating from therapeutic approaches, operate on pre-neoplastic cells, causing a stable arrest of cell proliferation [21,137]. This represents an important and powerful mechanism of tumor suppression which, in time, can be counterproductive due to the coexistence of senescent cells with SASP activity within the tumor and its microenvironment [49,148,149,150]. A microenvironment where senescent cells and their SASP co-exist is characterized by continuous modifications of the SASP over time, by the recruitment of immune cells by some mediators of the SASP, by the persistent inflammatory state, even though initially there are no cancer cells [149,151]. Any concomitant event that favors the onset of pre-neoplastic cells can be directed by senescent cells towards a more complete neoplastic transformation. In this case, the senescent cells can become pro-carcinogenic [21,49,148,149,151,152].

Today we know many aspects of the mechanisms by which senescent cells in the tumor microenvironment can antagonize or favor tumor growth [49,148,149,150].

### 3.1. Anti-Cancer Cellular Senescence

The anti-tumor activity of senescent cells is articulated on different types of interventions: recruitment of immune cells that can be decisive by exerting a strong and specific anti-tumor activity for tumor regression; secretion of thrombospondin 1 which prevents the senescent tumor cells to exit the state of senescence; clearance of pre-malignant cells pushed into cell senescence state that can be eradicated by immune cells recruited by some components of the SASP [21,151].

The effects of SASP on the tumor also seem to depend on the tumor stage: indeed, in precancerous tissues, the effects of SASP are mainly tumor-suppressive, based mainly on autocrine and paracrine cellular senescence, and on the induction of immunosurveillance [21,149,150,151]. However, in advanced tumors, SASP factors from senescent stromal cells such as cancer-associated fibroblasts (CAFs) can promote tumor growth [153].

Cellular senescence is a common feature in human neoplastic tissues [148,150]. Senescent cells have been well-documented in colon adenomas, which are precancerous lesions of invasive colon cancer [28]. Obviously, the presence of senescent cells in tumor tissues does not indicate whether their functional role is tumor-suppressive or pro-tumorigenic. Senescent cells are in a stable state of cell cycle arrest which represents a natural barrier to tumorigenesis. The tumor suppressive function of cellular senescence is due not only to the fact that the same pre-neoplastic cell that has become senescent no longer proceeds towards neoplastic transformation but also to the fact that in turn the senescent cell itself uses extrinsic mechanisms for this purpose. In fact, senescent cells are capable of promoting the senescence of adjacent cells, both by SASP and by cell-cell interactions, thus limiting the propagation of pre-malignant or fully malignant cells in their vicinity [21,49,148,149,150,151,152]. Furthermore, in certain situations, some factors of the SASP push the surrounding cells towards apoptosis or necrosis by means of TNF-α and ROS, or interleukin-6 (IL-6) [154].

Senescence has an important role in promoting immunosurveillance towards tumors: indeed, malignant senescent cells appear to be primed for clearance by the immune system [155].

### 3.2. Pro-Tumoral Cellular Senescence

There are numerous ways in which senescent cells can exert pro-tumorigenic activity [49,152]. In fact, some components of SASP have immunosuppressive activity and can also induce a direct stimulation of tumor proliferation, which can also be stimulated by extracellular vesicles [19,20,142,143,144,145,146]. Furthermore, some components of SASP can promote the migration and invasiveness of tumor cells and the onset of cancer stem-like cells [21,148]. It is clear that in the pro-tumorigenic or anti-tumor activity of cellular senescence, another variable is represented by the type of cell undergoing senescence, which can involve stromal and epithelial, normal, and tumor cells, and the type of cell affected by paracrine or juxtacrine effects in the microenvironment, which can be stromal, immune or tumor cells [156].

Another important aspect of the pro-tumorigenic action of SASP is the ability to promote angiogenesis [157].

The context-dependent effects of SASP also extend to its individual components, whereby a single SASP factor may be pro- or anti-tumorigenic, depending on the biological context. In fact, IL-6 in a normal environment helps to stabilize the senescent state, through an autocrine manner, while in an environment where there are tumor cells, it promotes the epithelial-mesenchymal transition favoring the spread of tumor cells [149,151].

An important role in the development and progression of tumors is also played by other cell types that are present within the tumor tissue, among which CAFs. CAFs are often senescent and associated with the SASP. In the tumor microenvironment, they are capable of promoting tumor expansion. Two types of inflammatory CAFs have been mainly identified in the tumor microenvironment: inflammatory CAFs, which are more tumor-promoting and with rich production of chemokines and cytokines, and myofibroblastic CAFs, which produce extracellular matrix but are less cancer-promoting. Both CAFs appear to act only on the surrounding cells. Senescent CAFs are present in the colon tumor microenvironment with active p38 signaling, which induces a pro-tumorigenic SASP [153,156,158,159].

Senescent cells and their SASP may contribute decisively to tumor progression and relapse by negative modulation of the immune system. In fact, CCL2 attracts a subset of CCR2+ myeloid cells that block natural killer (NK) cells. Furthermore, senescent cells present in the tumor stroma recruit myeloid suppressor cells CD11b+ and GR-1+ which switch off the cytotoxic activity of NK cells and CD8+ T cytotoxic lymphocytes [155].

## 4. Bacterial Toxins and Cellular Senescence

In the time course, bacteria have learned to use cellular senescence for their own purposes of replication and persistence of infection. In fact, the induction of senescence in immune cells makes their response less effective, favoring the persistence of the infection. Furthermore, senescence modulates the tissue microenvironment which may contribute to creating a favorable niche for subsequent reinfection, or simply a greater persistence of the infection [54,147].

It is clear that with advancing age there is a physiological increase in senescent cells, which accompanies the process of immunosenescence, and therefore a reduction in the effectiveness of numerous functional parameters, with a subsequent increased susceptibility to infections. One of the most important tools for bacteria to induce cellular senescence is bacterial genotoxins, which in most cases act by inducing DNA breaks that activate a molecular process of DDR [54,55,56,147]. However, if the DNA damage is too extensive, the cell either dies by apoptosis or acquires the state of senescence [53,54,55,56,147]. Of course, if the genotoxic damage is repairable, the cell recovers to normality and continues to survive [53].

### 4.1. Mechanisms of Cellular Senescence Induced by Bacterial Toxins

The most studied bacterial genotoxins inducing senescence are [54,55,56,147]:(1)Colibactin is produced by the B2 lineages of enteropathogenic *Escherichia coli* which causes DNA replication stress and induces DNA inter-strand cross-links. Cells that are able to survive such stress become senescent [160,161];(2)Cytoskeletal distending toxins (Cdts) are a family of toxins encoded by various Gram-negative pathogenic bacteria. Cdts are characterized by the Cdtb subunit which has structural and functional homology with DNase-I. This subunit induces DSB and SSB in DNA activating a strong DDR which ultimately leads to cell cycle arrest and elongation of cell morphology of senescent cells [55,56,147];(3)Cytotoxic necrotizing factor 1 (CNF1) is produced by extraintestinal pathogenic *E. coli* (B2 and D) and other Gram-negative bacteria, responsible for urinary tract infections, meningitis, and septicemia. CNF1 activates the Rho-GTPase family, blocking mitosis and cytokinesis, and inducing endoreplication and polyploidization. It is also classified as a cyclomodulin because it perturbs the cell cycle since it prevents CDK1-cyclin B-dependent cell cycle progression by arresting cells in the G2/M phase of the cell cycle [162,163]. While the purpose of its action is to destroy the epithelial barrier and facilitate efficient colonization, CNF1 counteracts apoptosis and reprograms the fate of surviving cells towards cellular senescence. However, these surviving cells are characterized by increased genomic instability [53,54,56].

Epidemiological data associate it with a higher incidence of IBD and CRC [43,44,45,46,47,48,163].

Genotoxin-induced cellular senescence, which in some cases has the characteristics of premature senescence, constitutes the rapid activation of an anti-tumor response in potentially oncogenic cells due to extensive DNA damage [55,56,147].

The ability of genotoxins to induce DNA damage which then leads to the survival of senescent cells with genetic instability opens up the problem of the carcinogenic potential of these toxins. However, there is no data demonstrating a correlation between these infections and the development of tumors over long periods of years. Certainly, in experimental models in vivo, it is possible to document a progression towards neoplastic transformation induced by genotoxins in a relatively short time [54,56,161]. For the long-time effects, it is necessary to assume that the senescent cells induced by the genotoxins survive for years so that the long time periods will allow an increase in the likelihood that the DNA lesions favor a neoplastic progression, in addition to other events that push the tumorigenic potential of these cells towards tumorigenesis. However, it is important to consider the fact that not only senescent cells induced by genotoxins can undergo a neoplastic transformation but due to their long survival with an active SASP, these can over time push the pre-neoplastic cells generated in the microenvironment towards neoplastic progression.

### 4.2. Mechanisms of Cellular Senescence Induced by TcdB

It is worth noting that TcdB also induces senescence in EGCs that survive the action of TcdB [11], which predominantly induces death by apoptosis/necrosis [10,23]. This phenomenon is of considerable importance given the increasing diffusion of CDI, the possible recurrences, and the correlations between CDI and IBS and IBD [69,70,71,73,74,125]. TcdB-induced senescence in EGCs is described by the following markers: positivity for SA-β-Gal, flat morphology, early and persistent DNA damage, persistent and irreversible cell cycle arrest due to alterations in expression and activity of cell cycle inhibitors and CDKIs, overexpression of sirtuin2 and sirtuin3 [11]. TcdB-induced EGC senescence is dependent on Rac1 glucosylation, c-Jun N-terminal kinase (JNK), and protein-kinase B o PKB (AKT) activation but independent of the p16 and p53/p21 pathways as well as ROS production [11].

In particular, TcdB-induced senescence in EGCs is caused by DNA damage as demonstrated by early phosphorylation of the kinase ATM and early and persistent formation of phosphorylated Histone 2AX foci but without SAHF formation [11]. Regarding persistent and irreversible cell cycle arrest observed in TcdB-induced senescence, it is independent of the senescence-inducing pathways p16/pRb and p53/p21 [11]. Indeed, although TcdB induces pRb hypophosphorylation, p16 expression levels do not change and p21 expression was increased only when EGCs have already acquired senescent phenotype, then p21 is involved only in the maintenance of the senescence state but not in induction as reported in other models [54,55,56,147,160]. Regarding the PTEN/p27 senescence-inducing pathway, the p27 plays a central role in TcdB-induced EGC senescence, as indicated by its early and persistent strong increase, whereas PTEN seems not involved because its expression is not modified [11]. In TcdB-induced EGC senescence the early and irreversible exit from cell cycle was mediated by: (1) early and persistent p27 overexpression, with early and persistent downregulation of cyclin B1, inactivation of CDK1, leading to inactivation of CDK1-cyclin B1 complex and an accumulation of the inactive hyperphosphorylated form of CDK1 reliable with the G1 and G2 arrest; and (2) early and persistent downregulation of c-myc which may fallow p27 upregulation [11].

Furthermore, the sirtuins, sirtuin2, and sirtuin3, seem to contribute to the maturation, maintenance, and stabilization of the senescence state, by controlling cell cycle progression but above all by antagonizing the lethal effects of high levels of ROS through an antioxidant response mediated by manganese superoxide dismutase, thus favoring the survival of senescent cells [11].

Finally, TcdB-induced senescence is characterized also by an upregulation of JNK and AKT which contribute to the survival of EGC after TcdB treatment and determining the choice between apoptosis or senescence [11].

Overall, induction of senescence by TcdB in EGCs shares some important markers with senescence induced by other bacterial toxins [54,55,160,164,165], such as persistent DNA damage and ROS production. However, differently from other bacterial toxins, ROS is not involved in senescence induction. In addition, in our model cell cycle arrest occurs both at the G1 and G2 phases, and the permanent arrest of cell cycle is not executed by p16 and p21, but by p27 [11].

To date, there is no in vivo demonstration of cellular senescence during CDI, although studies on the effects of Tcds in the tissue of mice during CDI demonstrated changes in the expression of cell cycle inhibitors and CDKIs indicative of cell cycle arrest [24,25], strongly suggesting that cellular senescence could occur during CDI in vivo.

## 5. Cellular Senescence and CRC

Senescent cells are acquiring an increasingly important role both in the pathology and prognosis of CRC [27,29,49,50,51,140,166,167,168,169,170]. The initially protective anti-tumor effects can become pro-carcinogenic over time and with changing conditions.

Senescent cells with SASP influence tumor progression, as described in immunosurveillance of tumor lesions [148,149]. This aspect is very relevant since CRCs are associated with high morbidity and mortality and account for 10% of cancer cases and 9.4% of deaths [38]. In patients with CRC, the absence of senescent cells or a high number of senescent cells are associated with a poor prognosis, underlining how a high concentration of senescent cells favors tumor progression [52,96,140,168,169,170]. This accumulation of senescent cells in the neoplastic tissue is probably due to inefficient anti-tumor immunosurveillance, also evidenced by the inability to clear senescent cells. Furthermore, the senescent cells accumulated within the tumor tissue create an excessive inflammatory environment that contributes to tumor progression [140].

It is possible that the measurement of cellular senescent cells in CRC tissue may become a predictive parameter in CRC [140,168,169,170]. However, cellular senescence is also used by the same tumor cells to inhibit some fundamental aspects of the immune response. It is also possible that senescence in colon tumor cells may be induced by senescent cells originating from other stimuli, such as CDI, which is induced by SASP. In this regard, an interesting mechanism has been described by which senescent tumor cells prevent CD8+ T lymphocytes from infiltrating the tumor: among the components of the SASP produced by senescent tumor cells of subjects with CRC, there is chemokine (C-X-C motif) ligand 12 CXCL12/colony-stimulating factor 1 (CSF1), which plays a primary role in the exclusion and activation of CD8+ T lymphocytes. CXCL12 overexpressed in senescent CRC tumor cells inhibits CD8+ T lymphocyte infiltration by reducing the production of T cell-attracting chemokines [29,169]. CSF1 can induce macrophage recruitment and induce polarization in M2 macrophages, which promote neoplastic development by inhibiting CD8+ lymphocyte activity and producing immunosuppressive cytokines such as IL-10 and TGF-β, which inhibit effective anti-tumor response, and factors such as Vascular-Endothelial Growth Factor, which promotes angiogenesis and matrix remodeling enzymes which modify the extracellular matrix and thus promote tissue remodeling, in turn enhancing tumor growth and invasiveness [166,169]. Of interest, a mouse model of CRC animal treated with neutralizing antibodies against CXCL12 and CSF1 showed a strong increase in CD8+ T lymphocyte infiltration and reduction in tumor growth [166,169].

Therefore, senescent tumor cells that can also be formed by the action of SASP form a protective barrier that inhibits the infiltration of CD8+ T lymphocytes and their activation. Furthermore, senescent tumor cells express a variety of SASPs that serve as a shield and promote a microenvironment that favors tumor growth [29,169].

## 6. CDI and Cellular Senescence

If CDI and its recurrences can represent a risk for the induction of CRC, it is important to identify the possible underlying mechanisms, bearing in mind that between the initial event of infection and the appearance of CRC a long-time span may be needed. However, in recent years some evidence has accumulated, and we have many important elements, though indirect, that suggest what could be this pathway.

### 6.1. The Environment of the CDI

Over the course of CDI, TcdB reaches different concentrations at various sites of *C. difficile* growth, particularly as the infection deepens into the mucosa. Therefore, it is foreseeable that both epithelial cells and submucosal cells come into contact with doses of TcdB which do not induce death by apoptosis and/or necrosis [5,9] but instead activate a process of cellular senescence [11]. As reported above, the key event of senescence activation is cell cycle arrest, because the arrest of the cell cycle and cell proliferation is a common and early event induced both by TcdA and TcdB in vitro and in vivo [5,6,10,11] and usually all cell types under appropriate stimuli are capable of go into senescence [12], it is likely that all cell types susceptible to Tcds activity that survive the Tcd effects may go into cell senescence both in vitro and in vivo.

Cellular senescence could also involve the various types of immune cells recruited by the inflammatory process and also the cells of innate immunity residing in the colonic mucosa. An important factor in the cellular senescence induction process induced by TcdB is the inflammatory response characterized by the presence of CKs: on one hand, these synergize with TcdB in enhancing its cytotoxic action [10,23], and on the other, the cytokines by themselves can contribute to promoting senescence [157] in the cells surviving to TcdB. Therefore, although the CKs operate a strong selection on the surviving cells, they also can contribute to their transformation into senescent cells. Furthermore, the cytokines produced during the CDI could induce senescence even in cells unaffected by the TcdB.

The generation of senescent cells during CDI is also due to tissue regeneration process occurring once the infection has been overcome, further tissue regeneration could greatly benefit from senescent cells induced by TcdB and cytokines, which will contribute with their SASPs to the formation and control of new reparative and functional tissue [15,20].

In conclusion, a wave of senescent cells may be formed during CDI, with the contribution of TcdB, cytokines, and the reparative process. Furthermore, as demonstrated in other infections, other structural and metabolic components of *C. difficile* may be involved in pushing cells towards senescence. To date, there are no in vivo data on the presence of senescent cells after CDI, as demonstrated in both animal and human models with other types of bacterial infections [53,54,55,56,147,160,162,163,164,165].

However, it is possible to hypothesize a path that starting from CDI can lead to colon carcinogenesis (Figure 3).

### 6.2. CDI and IBS Correlation

Among the main pathophysiologic factors involved in IBS, there is substantial documentation of a low-grade inflammation harbored in the gut of these individuals [171]. There is presently substantial evidence in the literature that IBS onset often follows a bacterial infection (“post-infectious IBS”) [172] and that CDI plays a substantial role in this topic [125]. In fact, microbial antigen presentation to the mucosa may induce, in a genetically primed host, immune gut activation with low-grade intestinal inflammation and subsequently neuronal structural and functional alterations; these, in turn, cause regional intestinal hypersensitivity and motor dysfunction [173]. The immune gut activation and the low-grade inflammation found in these subjects are mainly represented by an increased number of mast cells and CD3+ T cells in colonic biopsies of IBS patients compared to controls [174]. This low-grade inflammation can induce senescent cells [28] (Figure 3a).

Therefore, in subjects in which IBS has developed following CDI, we can hypothesize that IBS is due to the impairment/decrease of cells such as EGCs and neurons, responsible for an altered functionality also increased by senescent cells induced during the CDI [172] (Figure 3b).

Thus, if after a CDI the subject develops IBS, this might have different characteristics from an IBS developed for other causes, because after CDI the IBS presents a greater quantity of senescent cells generated by the infection, which add up and interact with those generated by IBS itself (Figure 3).

Based on these considerations, in post-CDI IBS there is likely a concomitance of functional alterations and significant accumulation of senescent cells that have been able to survive for a long time due to the contribution of the low but continuous level of inflammation. It is also predictable that IBS after CDI may display greater severity in terms of impaired functionality and inflammation, also due to senescent cell SASP and concomitant dysmicrobism, even though to date there is no such demonstration in the literature. Therefore, the appearance of pre-neoplastic cells over time will find an environment favorable to tumor progression and it is possible to hypothesize a correlation between CDI, IBS, and CRC (Figure 3).

### 6.3. CDI and IBD Correlation

IBD are chronic inflammatory intestinal condition including two entities, ulcerative colitis, and Crohn’s disease, with no exact known cause [175]. These conditions are characterized by a variable inflammation status of the intestinal wall, mucosal or transmural, leading to several clinical manifestations including bloody diarrhea, abdominal pain, intestinal (sub)occlusion, fibrosis, fistulization, etc. [176]. Due to abnormal immunity patterns and the frequent use of immunosuppressive therapies, IBD patients are at risk for and often develop CDI [73,74,177,178]. Of interest, patients with IBD have a well-defined overlap (especially during remission) with IBS [179] and a substantially increased risk of CRC [30,31,32,33].

In subjects with IBD, there is a persistent state of inflammation with peaks with continuous processes of tissue repair and consequently of tissue stress. In these conditions, the formation of senescent cells is due to the pathological stress of the cells, the strong inflammatory response, and to a strongly altered microbiome that contributes to the inflammatory state [27,28,29,52] (Figure 3a).

Initially, cell senescence may act as a barrier to carcinogenesis, but this barrier decreases in dysplastic lesions. In this context, a very important role is played by the cells of the innate immune system [175], which by their subtypes can favor the persistence of the inflammatory state and, above all, can be particularly unable in eliminating the senescent cells that are generated.

Thus, an individual with IBD who has had a prior CDI followed by IBS may enter a path with a progressive enrichment and accumulation of senescent cells deriving from the three conditions (Figure 3b). Therefore, the likelihood of developing a neoplastic progression might be enhanced. Other factors can also contribute to this neoplastic progression; the profound alteration of the microbiome often present in IBD patients not only favors the inflammatory response contributing to the generation of new senescent cells but alters the various types of immune cells and the homeostasis between colon cells and the microbiota.

Furthermore, the continuous reparative process also further contributes to the generation of senescent cells. However, the key event is the presence of numerous senescent cells, which are stimulated by inflammatory cytokines. These cells, synergically to some SASP factors in a profoundly altered immunological and microbiological environment with a reduced ability to eliminate senescent cells, are more likely to push the more numerous pre-neoplastic cells towards neoplastic progression in CRC (Figure 3).

This exemplified picture becomes even more complex if, in all the above situations, we take into consideration CDI recurrences, likely to further increase the incidence of CRC (Figure 3b). In fact, it is clear that in the above-described subjects reinfections with *C. difficile* increase the probability of neoplastic transformation to CRC. Furthermore, in these situations, the potential aggravating contributions to neoplastic progression due to infections with *E. coli* and other bacteria must be also taken into account. These bacteria, by producing genotoxins, can generate genetically unstable senescent cells, a target of the pro-tumorigenic action of SASP of the senescent cells generated in the various phases of pathological progression.

## 7. CDI and Cellular Senescence and CRC

As above reported, senescent cells are acquiring an increasingly important role in CRC [27,29,49,50,51,140,166,167,168,169,170]. The initial protective anti-cancer effects of senescent cells can become pro-tumoral over time and with changes in the microenvironment and SASPs.

It has been reported, as described above, that the absence or a high number of senescent cells in patients with CRC are associated with a poor prognosis, highlighting as senescent cells at high concentrations favor tumor progression [52,96,140,168,169,170]. This accumulation of senescent cells in the neoplastic tissue is probably due to inefficient anti-tumor immunosurveillance, and also due to the inability to clear senescent cells. Furthermore, the senescent cells accumulated within the tumor tissue can create an excessive inflammatory environment that contributes to tumor progression [140].

Then, senescent cells generated by one or more CDIs together with senescent cells generated by IBS and by IBD can alter the balance between the anti-tumoral and pro-tumoral effects of senescent cells and favor CRC progression (Figure 3).

Moreover, senescent tumor cells formed by the action of SASP following infections such as CDI could form a protective barrier that inhibits the CD8+ T lymphocyte infiltration and their activation. In addition, the accumulation of senescent tumor cells in subjects developing IBS and IBD after CDI could strongly increase the expression of a variety of SASPs that act as a protective shield against CRC so promoting a microenvironment strong conductive to tumor growth [29,169].

## 8. Conclusions

Cellular senescence has been reported in several gastrointestinal diseases including IBS, IBD, and colorectal carcinogenesis. In subjects with IBS, indeed, the low-grade inflammation may induce senescent cells. In subjects with IBD, there is a pathological stress of gastrointestinal cells due to the strong/persistent inflammatory response and to a strongly altered microbiome which contributes to the inflammatory state and continuous processes of tissue repair, which altogether lead to the formation of senescent cells. Therefore, given that CDI generates senescent cells by itself, and favors the development of IBS and IBD, CDI can contribute to increasing the load of senescent cells in these two pathologies and, in IBD patients, to increase the incidence of CRC. The increasing diffusion of CDI worldwide due to the unstoppable spread of *C. difficile* in the anthropized environment and the progressive human colonization, the possible recurrences, the correlations between CDI and IBS and IBD (where the latter two may induce cellular senescence for their intrinsic pathologic mechanisms), and taking into account also that CDI occurs mainly in aging people where there is a physiological increase in senescent cells for the advancing age, could cause an accumulation of senescent cells which over time may have important functional consequences and contribute to the development of CRC.

Epidemiological data of senescence induced by bacterial genotoxins associate it with a higher incidence of IBD and CRC.

Several pieces of evidence indicate that senescent cells are acquiring an increasingly important role both in the pathology and prognosis of CRC. Indeed, it has been reported that in patients with CRC, a high number of senescent cells are associated with a poor prognosis, underlining how a high concentration of senescent cells favors tumor progression. The senescent cells initially can play a protective anti-tumor effect, but this can become pro-carcinogenic over time and with changing conditions. In fact, when there is a low number of senescent cells this can mediate tumor suppression by several mechanisms among which the main ones are the arrest of the cell cycle and the secretion of specific factors and cytokines by SASP. On the contrary, in conditions where there is an accumulation of senescent cells (e.g., chronic damage, deregulation of immunosurveillance, aging) these could contribute to tumor progression.

Therefore, after CDI, particularly in subjects who develop IBS and/or IBD, the strong accumulation of senescent cells with long survival and the inability of the immune system to eliminate all the senescent cells generated could push pre-neoplastic cells generated in the colon towards the complete neoplastic transformation in CRC by SASP and immunosuppression/immune evasion. Pre-neoplastic cells could thus appear as a result of various pro-carcinogenic events, such as infections with bacteria that produce genotoxins which generate cells with high genetic instability.

The therapeutic goal for CRC would be to eliminate senescent cells (senolysis) and/or neutralize the SASP. While awaiting such approaches, we must currently take into account the need to monitor subjects following CDI, especially those affected by IBS and/or IBD, and exert active surveillance with particular attention to CDI recurrences, to prevent the potential increased risk of progression toward CRC.

## 9. Future Prospects

From a mere biological phenomenon, cellular senescence has progressively become an important pathophysiological mechanism, assuming a key role in most pathological conditions, often with opposite effects on the pathological process depending on the conditions to be defined. The better knowledge of senescence, therefore, paves the way for possible interventions to reduce the pro-pathological effects.

CDI, given its prevalence and recurrence rate, the ability of TcdB to induce senescent cells, and the direct correlation between CDI, IBS, and IBD that could cause an accumulation of senescent cells with important functional consequences, requires close clinical and laboratory monitoring for the detection of senescent cells or SASP release over time due to the higher incidence of IBS, IBD, and CRC in subjects who have had CDI.

When it will be possible to demonstrate in vivo in these subjects the presence of senescent cells or the secretion of molecules characterizing SASP, the progress that we expect on senolytic or anti-SASP therapies could radically change the course of the various steps leading to the induction of CRC.

## Figures and Tables

**Figure 1 ijms-24-08155-f001:**
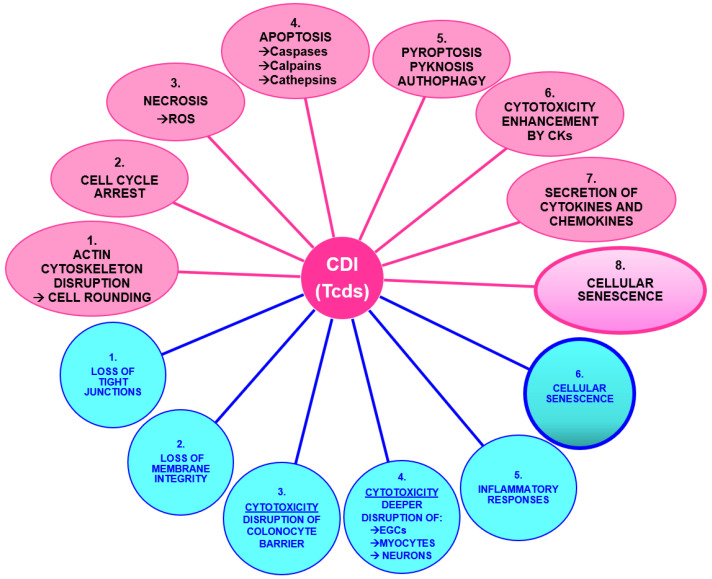
Effects of Tcds in vitro and in vivo. In pink the in vitro effects and in cyan the in vivo effects. Abbreviations: *Clostridioides difficile* (*C. difficile*) infection (CDI); *C. difficile* toxins (Tcds); reactive oxygen species (ROS); tumor necrosis factor-alpha (TNF-α) plus interferon-gamma (IFN-γ) (CKs).

**Figure 2 ijms-24-08155-f002:**
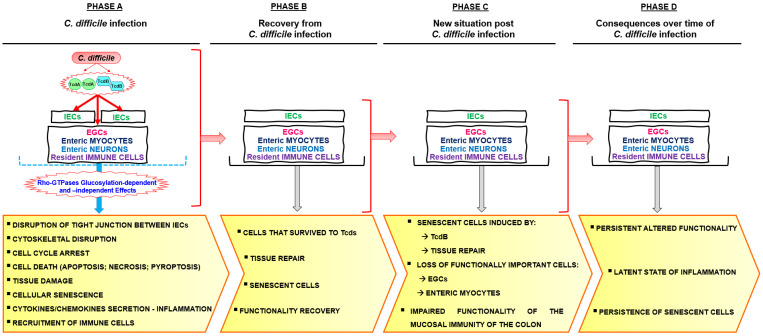
*C. difficile* infection and functional consequences. Abbreviations: *Clostridioides difficile* (*C. difficile*) toxins (Tcds); *C. difficile* toxin A (TcdA); *C. difficile* toxin B (TcdB); intestinal epithelial cells (IECs); enteric glial cells (EGCs).

**Figure 3 ijms-24-08155-f003:**
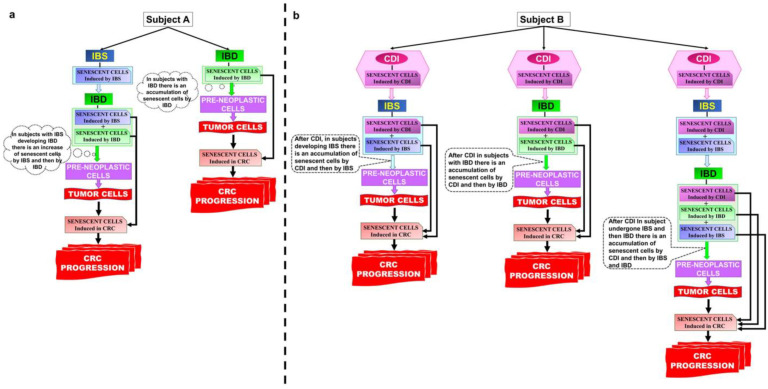
Role of senescent cell cargo in contributing to the development of CRC. (**a**) Represent the pathway of senescent cell accumulation and the development of CRC in subjects developing IBS and IBD without CDI; (**b**) Represent the pathway of senescent cell accumulation and the development of CRC in subjects developing IBS and IBD after CDI. Abbreviations: *C. difficile* infection (CDI); irritable bowel syndrome (IBS); inflammatory bowel diseases (IBD); colorectal cancer (CRC).

## Data Availability

Not applicable.

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
