# Peer review of "Clostridioides difficile Toxin B Induced Senescence: A New Pathologic Player for Colorectal Cancer?"

_ijms, 2023, doi:10.3390/ijms24098155_

Round 1

Reviewer 1 Report (New Reviewer)

The manuscript presented by Katia Fettucciari and collaborators is a comprehensive review concerning TcdB toxin produced by Clostridioides difficile as potential factor to develop colorectal cancer. I very much enjoyed the manuscript, it incorporates substantial information regarding epidemiology of C. difficile, mechanism of action of toxins produced by C. difficile and cellular senescence related with the toxins. All of the studies that I expected to see discussed were and I could not identify any serious flaws in the arguments presented. In my opinion the manuscript is complete and suitable for publication.

Author Response

Please write down "Please see the attachment." 

Reviewer 2 Report (Previous Reviewer 2)

The paper is focuses on the Clostridioides difficile toxin b induced senescence: a new pathologic player for colorectal cancer? Topic could be considered interesting, but the manuscript must be much better developed. However, extensive revision is required before any considerations about publication. Regarding the manuscript I have the following main concerns/suggestions:

Title is also about colon cancer. A separate paragraph must be developed in the Introduction, related to this condition, making a clear frame for it (prevalence, mortality, techniques of investigation, treatment management, etc.). In this regard, I suggest checking and referring also to Pallag, A.; et al. Monitoring the effects of treatment in colon cancer cells using immunohistochemical and histoenzymatic techniques. Rom. J. Morphol. Embriol., 56(3), 2015, 1103-1109. PMID: 26662146.

Moreover, which are the risk factors for CDI? Relevant, recent papers must be consulted - PMID: 32967323

last paragraph of Introduction. Aim of the study must be clearer presented and developed. What is the novelty/special aspects your study brings to the field. Why have you chosen this topic? Give a reason to increase the interest of the readers in reading your paper, as there are hundreds of papers in similar topic.

It must be added a 2nd section of Methodology for literature search/selection or a similar title. It would therefore be advisable to present the methodology for selecting bibliographic resources (databases used and the reason for choosing those data basis, types of documents, filtering results, inclusion/exclusion criteria for manuscripts: language, key words, duplicates, etc.). Moreover, have you searched (graphically) the impact of the topic on the general literature? In this regard I suggest using WoS. Providing a figure would be relevant and will improve the graphical part of your research which consist in only one figure.

If this is a Review, I do not understand how very recent papers in the field have been omitted, considering literature searching that must be a complex and rigorous one. (i.e., related to the CDI epidemiology - https://doi.org/10.3390/su12114439  ). Epidemiology section must be also improved with at least one table, summarizing studies from different countries, to make a clear image of this disease. Last column should be Ref. (references). Tabulated part is missing in your paper.

Subsection 7.2. It is not clear the role of gut in IBD. Please describe. https://doi.org/10.3390/diagnostics11061090

Figure 1 is blurred, to unreadable. Provide a best quality figure. Print screen the original figure, and paste. Do not save it in any other format, as it will lose clarity.

L721. Section of Abbreviation must be removed. All abbreviations must be inserted in the main text. Abbreviations must be explained when they first appear in the main text, even if they have been included in the abstract, and contribute to making the text easier to read and the information conveyed more efficiently. Once an abbreviation has been established and explained, it will be used throughout the entire manuscript, except for the abstract, where it must be treated separately. Check the Instructions for authors: Abbreviations should be defined the first time they appear (added in parenthesis, after the written-out form) in each of the three sections: the abstract; the main text; and under the first figure or table. When defined for the first time, the abbreviation should be added in parentheses after the written-out form”. The Instructions are given to be applied, not being optionally.

Check all self-citations if you really need all of them (one author have 8).

Author Response

Point 1: The paper is focuses on the Clostridioides difficile toxin b induced senescence: a new pathologic player for colorectal cancer? Topic could be considered interesting, but the manuscript must be much better developed. However, extensive revision is required before any considerations about publication. Regarding the manuscript I have the following main concerns/suggestions:
Title is also about colon cancer. A separate paragraph must be developed in the Introduction, related to this condition, making a clear frame for it (prevalence, mortality, techniques of investigation, treatment management, etc.). In this regard, I suggest checking and referring also to
Pallag, A.; et al. Monitoring the effects of treatment in colon cancer cells using immunohistochemical and histoenzymatic techniques. Rom. J. Morphol. Embriol., 56(3), 2015, 1103-1109. PMID: 26662146.
Point 2: Moreover, which are the risk factors for CDI? Relevant, recent papers must be consulted - PMID: 32967323

Point 3: last paragraph of Introduction. Aim of the study must be clearer presented and developed. What is the novelty/special aspects your study brings to the field. Why have you chosen this topic? Give a reason to increase the interest of the readers in reading your paper, as there are hundreds of papers in similar topic.
It must be added a 2nd section of Methodology for literature search/selection or a similar title. It would therefore be advisable to present the methodology for selecting bibliographic resources (databases used and the reason for choosing those data basis, types of documents, filtering results, inclusion/exclusion criteria for manuscripts: language, key words, duplicates, etc.). Moreover, have you searched (graphically) the impact of the topic on the general literature? In this regard I suggest using WoS. Providing a figure would be relevant and will improve the graphical part of your research which consist in only one figure. If this is a Review, I do not understand how very recent papers in the field have been omitted, considering literature searching that must be a complex and rigorous one. (i.e., related to the CDI epidemiology - https://doi.org/10.3390/su12114439 ). Epidemiology section must be also improved with at least one table, summarizing studies from different countries, to make a clear image of this disease. Last column should be Ref. (references). Tabulated part is missing in your paper.
Point 4: Subsection 7.2. It is not clear the role of gut in IBD. Please
describe.
https://doi.org/10.3390/diagnostics11061090

Author Response to the report of
Reviewer 2 point 1-4: The Report of Reviewer 2 does not represent the scientific objectives that you intended to achieve so we does not done the request at point 1-4. In fact, the Reviewer requests a transformation of our comprehensive narrative Review in systematic/meta-analysis Review, and this does not represent the scientific objectives that we have intended to achieve. Furthermore, the Reviewer 2 requests an enlargement of the References on insights and areas which are already wide described in our Review and cited and more important ask of insert 4 references of the Reviewer of his research team.

Point 5: Figure 1 is blurred, to unreadable. Provide a best quality figure. Print screen the original figure, and paste. Do not save it in any other format, as it will lose clarity.

Point 6: L721. Section of Abbreviation must be removed. All abbreviations must be inserted in the main text. Abbreviations must be explained when they first appear in the main text, even if they have been included in the abstract, and contribute to making the text easier to read and the information conveyed more efficiently. Once an abbreviation has been established and explained, it will be used throughout the entire manuscript, except for the abstract, where it must be treated separately. Check the Instructions for authors: Abbreviations should be defined the first time they appear (added in parenthesis, after the written-out form) in each of the three sections: the abstract; the main text; and under the first figure or table. When defined for the first time, the abbreviation should be added in parentheses after the written-out form”. The Instructions are given to be applied, not being optionally.

Response to the report of
Reviewer 2 point 5,6: Done.

Point 7: Check all self-citations if you really need all of them (one author have 8).

Response to the report of
Reviewer 2 point 7: In any of the self-citations inserted in the Review it is described a different and relevant aspect of C. difficile pathogenesis and underlining mechanisms then we really need of all self-citations

Reviewer 3 Report (New Reviewer)

The authors of the work "Clostridioides difficile toxin b induced senescence: a new pathologic player for colorectal cancer" carry out a great bibliographic review work on CDI and its possible relationship with CRC. The work is developed with a logical structure and the statements are supported by appropriate references. In fact, the authors use a total of 173 references, some of which are very current. However, I would like to make a number of comments:

1. Due to the large amount of data provided, the generation of figures summarizing the effects of Clostridium difficile infection would be helpful, in addition to Figure 1.

2. I also think that a section on "future prospects" would help to complete this very good review.

3. Is the CDK1 appointment repeated on line 75?

4. Regarding the references, some changes must be made:

a. References 6, 7, 8, 9, 22, 36, 52, 53, 56, 57, 58, 63, 81, 86, 89, 96, 118, 119, 120, 127, 161, 162, 164 and 170 have not the year in bold.

b. Are references 22, 56, 57, 58, 86, 89 and 161 complete? They seem to be missing the volume and/or page numbers.

c. The pages in the references are indicated with different formats. Please homogenize.

d. Species names should be written in italics (example: reference number 150).

Round 2

Reviewer 2 Report (Previous Reviewer 2)

Revision poorly done (nothing related to the content gaps was completed). The topic, in this shape, has no relevance.

Author Response

This manuscript is a resubmission of an earlier submission. The following is a list of the peer review reports and author responses from that submission.

Round 1

Reviewer 1 Report

Appreciate to submit review manuscript to International Journal of Molecular Sciences. The severity of Clostridioides difficile infection (CDI) has been considered as a critical inducer of diseases including IBS, IBD, and CRC. In this manuscript, the authors demonstrated well why senescent cell is essential to be monitored in CDI and how dangerous it is. However, there are several parts needed to be corrected or added for better understanding. 

1.     Page 2, line 58 : The percentage that C. difficile colonizes in healthy infants ranges from 17-70%. As the range seems huge, it would be good to elaborate on the cause of it. 

2.     Page 4, line 153 : Typo

3.     Page 4, line 174-177 : The authors mentioned about a loss of tight junction resulting in decreased membrane integrity. Compared to the other mechanisms, it looks too simple. The authors would better elaborate on it precisely (e.g. which mechanisms are engaged in ZO-1 decrease). In addition, immune cells recruitment is induced not only by tight junction dysfunction but also cytokines derived from epithelial cells. It should be also described more. 

4.     Page 4-7, line 184-323 : This part is about markers of senescent cells. But it does not describe well to show the engagement of those markers on CDI. As this review paper is not for the only senescence mechanisms. The authors would better consider to delete this part. 

5.     Page 8-10, line 388-479 : Same with comment 4. The authors dealt with anti-cancer, pro-tumor properties of senescent cells, which is not directly related with CD. 

6.     Overall, several parts have leaps in the manuscript, which make us think it is not solid. 

Reviewer 2 Report

The current review article discusses the role of Clostridium difficile in digestive system pathological processes. The topic is relevant and interesting, but the poor organization of the information, the poor understanding of the information, the lack of correlation with the requirements of the journal, and the incorrect use of bibliographical references suggest that it was done in a hurry.

Please check the instructions for authors and the template provided by the journal regarding the organization of the title. There is no need for a title written in full capital letters. Please revise the manuscript.

Please check the instructions for authors and the template provided by the journal regarding the use of keywords. There is no need for abbreviations in this section. Please revise the manuscript.

Please check the instructions for authors and the template provided by the journal regarding the organization of the chapters. The present review article should start with a first chapter entitled "Introduction’’, without the use of subchapters, and in the last paragraph of the introduction, the aim of the paper should be clearly stated, along with motivations for the contributions made to the field and aspects regarding the novelty of the approach.

L50; L68- Too many bibliographic resources for such a short phrase. Please remove the unsignificant ones.

L75- the abbreviations are not used in this format (MDR - Multidrug Resistant). Please check the instructions for the authors and revise the whole manuscript in terms of abbreviations. The abstract should be treated separately from the main text.

L92-L101- it is not accepted as an enumeration for which each term has the same bibliographical source; either only one date is put at the end of the list or other bibliographical resources are searched for each item, especially as multiple bibliographical sources have been used illogically in smaller paragraphs.

The Conclusions section needs to be improved because fewer things were highlighted than were evaluated in the present review.

The entire manuscript is a string of poorly organized information that is difficult for readers to follow and does not follow the rules of typesetting and organization imposed by the journal. Furthermore, numerous bibliographical references are misused, many with low relevance, or the same bibliographical references are applied to 2-3 consecutive paragraphs. It is recommended to clarify the information, use much better structuring of the data (tables, figures), and manage bibliographical references much better.

Reviewer 3 Report

The manuscript primarily addresses the association between Clostridioides difficile infection (CDI) and cell senescence, the direct link between CDI and conditions such as irritable bowel syndrome (IBS) and inflammatory bowel diseases (IBD), the alterations in senescent cells, as well as the role of cell senescence in both anti-tumor and pro-tumoral activity. However, a logical inconsistency is observed in the manuscript. I recommend rejecting it.

Major issues

1. The title of the review misrepresents its content. The reader is led to believe that the review will primarily discuss the induction of cell senescence and colorectal cancer by Toxin B. However, a substantial portion of the review, including sections 2, 4, 5 and 7, is devoted to cell senescence and is not directly related to Toxin B. The review (~70%) has content pertinent to cell senescence but not to the role of Toxin B in cell senescence. The present format of the review predominantly emphasizes cell senescence, rather than the intersection between Toxin B and cell senescence. It would be more appropriate to revise the review and present its information in a concise format that aligns with the title, rather than presenting it as a lengthy and divergent piece of literature.

2. line 491-494: It is not logically sound to infer that cells which survive the Tcd effect would undergo senescence based on the two events - the cells go into senescence and the arrest of the cell cycle by TcdB.

3. line 514-516: This speculation lacks logical justification.

Reviewer 4 Report

Good work! Please double-check the grammar and spelling carefully. Check if you have defined all the acronyms. 

Line 33 – Remove to after for

Line 51 - acquired CDI” and are in – are should be replaced with is

Line 55 - comorbidities as diabetes -should be “such as”

Line 59 – is that 4-15% of healthy adults?

Line 61 – are progressively increasing…

Line 95 – change spores to endospores as bacteria form endospores. It is different from spores.

Line 99 – hypervirulent RTs - define RTs.

Line 112 – TcdA and TcdB…

Line 142-144 – Sentence starts with “In fact….is not clear

Line 153 - ), ???